# Neuromorphic Edge Intelligence for Rural Environmental Monitoring

ATAKAN ARAL, Umeå University, Sweden and University of Vienna, Austria

Edge Intelligence is the fusion of Edge Computing and Artificial Intelligence. It promises responsiveness, privacy preservation, and fault tolerance by moving parts of the Artificial Intelligence workflow from centralized cloud data centers to distributed edge servers. However, most Edge Intelligence implementations so far are limited to urban areas, where the infrastructure is substantially more dependable. This work instead focuses on a class of applications involved in environmental monitoring in remote, rural areas such as forests and rivers. There are new challenges in such applications, including access to electricity and network. We propose Neuromorphic Computing as a promising solution to the energy, communication, and computation constraints in such scenarios and identify future research directions in Neuromorphic Edge Intelligence for rural environmental monitoring, namely distributed model synchronization, edge-only learning, and aerial networks.

Key Words: Edge Computing, Artificial Intelligence, Edge Intelligence, Rural Computing, Environmental Monitoring

## 1 INTRODUCTION

Environmental monitoring plays a crucial role in contributing to the United Nations Sustainable Development Goals, particularly toward *Clean Water and Sanitation*, *Climate Action*, *Life Below Water and on Land*, *Sustainable Cities and Communities*, and *Responsible Consumption and Production*. Internet of Things (IoT) technology enhances environmental monitoring systems by enabling data collection, transmission, and analysis from various sensors and devices. Driven by the exploding number of IoT devices and the amount of data generated at the edge of the network, Edge Intelligence (EI) [5] is widely considered the next logical step for real-time distributed data processing. EI arises from the convergence of Artificial Intelligence (AI) and Edge Computing (EC) and proposes utilizing the prevalent EC resources for training AI models and inferring based on thereof. Consequently, streaming big data from IoT devices can be processed in close proximity, which could bear various benefits, including bandwidth savings, high responsiveness, and privacy preservation. Initial use cases for EI, such as traffic control, smart factories, and smart cities, have been almost exclusively located in urban areas [5]. These environments are characterized by operational utilities (e.g., electric power) and high-bandwidth Internet access. However, we argue that EI can also find use in environmental monitoring applications targeted at rural and remote areas. In such a scenario, EI has to encounter also the challenges such as data insularity, low computation capability, and limited fan-in.

Another pressing issue that restrains large-scale processing of IoT data is energy consumption. Not to mention the current global energy crisis, electricity use of data centers is already a controversial topic. In 2019, the city of Amsterdam imposed a moratorium on building new data centers due to their high electricity budget, which was expanded to the whole country by the national government in 2022. ICT currently accounts for 5% to 9% of global electricity consumption with comparable carbon emissions to air travel. Relying on redundancy (of IoT devices, communication links, and processors) to withstand the data explosion would increase energy consumption exponentially, with estimates exceeding 20% of the global electricity demand by 2030 [2].

Our main contributions in this work can be summarized as follows. First, we present three classes of rural environmental monitoring applications (i.e., pollution monitoring, disaster warning, and industrial IoT) and outline common characteristics and challenges (Section 2). Then, we introduce EI as a promising solution to most of these challenges, which in turn has its own limitations (Section 3). Further, we discuss how Neuromorphic Computing (NC), a novel non-von Neumann technology, fits into the picture and identify future research directions for its practical use in

Author's address: Atakan Aral, Umeå University, Umeå, Sweden and University of Vienna, Vienna, Austria, atakan.aral@umu.se.

enhancing edge intelligence for environmental monitoring use cases in rural areas (Section 4). Finally, we present a numerical analysis to demonstrate the feasibility of one of these research directions, namely, distributed model synchronization (Section 5) and conclude the paper with final remarks (Section 6).

We define Neuromorphic EI as *a distributed computing architecture, where brain-inspired, massively parallel, and event-driven hardware is deployed at the edge of the network, close to IoT data sources.* There have been initial studies addressing rural computing in general, such as [6]. Recently, the convergence of EI and NC has also been introduced [10]. However, to the best of our knowledge, this work is the first to identify rural environmental monitoring as a new research direction and also the first to employ Neuromorphic EI for environmental monitoring. Therefore, we believe it will be highly beneficial for scientists and practitioners in environmental informatics alike.

## 2   RURAL ENVIRONMENTAL MONITORING

In this section, we first list the most widespread forms of monitoring rural environments. Then, we discuss the defining characteristics and limitations that distinguish them from urban monitoring applications, such as noise level, metropolitan air quality, heat island, and urban climate monitoring.

### 2.1   Use Cases

*2.1.1   Pollution Monitoring.* Pollution monitoring is a process that involves measuring the ambient level of pollution in various mediums. Increasing global human activity and its consequent impact on the environment make it crucial to monitor air and water quality in rural areas. Pollution monitoring is beneficial not only for detecting sudden events such as leakages and enabling countermeasures but also for long-term modeling, which helps environmental scientists better understand the trends, impacts, root causes, etc. Global Environment Monitoring System (GEMS) [13] by the United Nations Environment Programme is a comprehensive attempt at worldwide pollution monitoring. The system focuses on air and water quality monitoring through the combination of low-cost IoT sensors, remote sensing technology, and traditional monitoring methods.

In water quality monitoring, there exist monitoring systems for marine regions and freshwater bodies (both ground and surface water). SWAIN project [12], supported by the European Innovation Council CHIST-ERA programme, focuses on surface waters, particularly rivers. The project aims to detect and locate pollutant sources (e.g., industrial leaks or failed wastewater treatment plants) through an unprecedented implementation of EC and IoT for the real-time analysis of water contamination data. Timely decision-making is crucial in this use case as the river water polluted upstream might be used for irrigation or even for municipal water intake downstream. Such a scenario is illustrated by the Ergene River in Figure 1, where the potential health risk is apparent with highlighted heavy industrial development and irrigation areas. Ergene is one of the two use cases in the SWAIN project (the other being the Kokemäki River).

*2.1.2   Disaster Warning.* Environmental monitoring systems are also widely used to detect and predict natural and anthropogenic disasters. Large-scale deployment of seismic sensors enables effective monitoring of earthquakes (e.g., EMSC or USGS), volcanic activities, and avalanches. Intercantonal Measuring and Information System (IMIS) is the snow meteorology and avalanche warning network covering the Swiss Alps [15]. IMIS consists of snow stations with sensors for snow depth, air and surface temperature, etc., and wind stations measuring wind speed and direction. One of the most ambitious disaster warning infrastructures is maintained by the Comprehensive Nuclear-Test-Ban Treaty Office (CTBTO), which monitors the whole planet from more than 300 stations for signs of nuclear explosions. The International Monitoring System (IMS) [3] uses data from seismic sensors (to monitor shockwaves), hydroacoustic

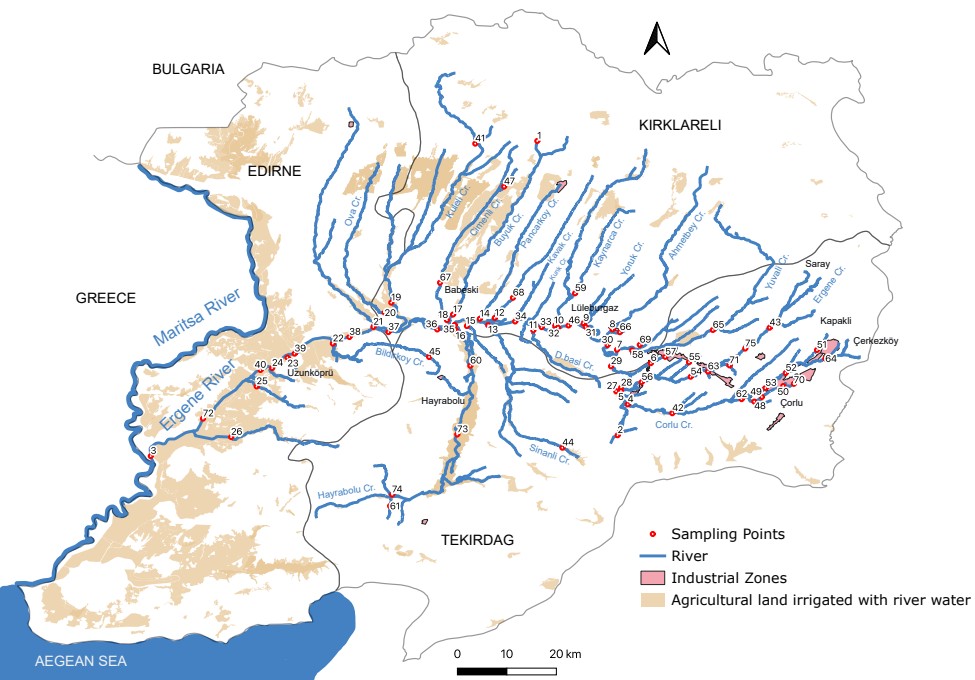

Fig. 1. Geographical Overview of the Ergene Watershed Located in Northwestern Turkey as a Water Quality Monitoring Use Case (Image Courtesy of TUBITAK Project 115Y064).

sensors (sound waves in the oceans), infrasound sensors (ultra-low frequency sound waves), and radionuclide sensors (radioactive particles in the atmosphere). Collected data is transmitted to the IMS data center in Vienna, Austria, and processed to detect nuclear explosions, which potentially violate the Comprehensive Nuclear-Test-Ban Treaty adopted by the United Nations.

*2.1.3    Industrial IoT.* The third and final class of rural environmental monitoring applications that we discuss is the industrial IoT systems. Although industries are often located within or near urban areas, e.g., in industrial zones, there exist at least two industrial use cases that are inherently rural. In the agricultural industry, IoT-driven smart farming is increasingly prevalent. Here, sensors collect continuous data, including light, humidity, temperature, and soil moisture, to model crop health, yield, etc. Furthermore, actuators can automate various tasks, including seeding, seedling, pollination, fertilization, irrigation, and harvesting. The second use case concerns the oil and gas industry, particularly well monitoring systems. Rural oil and gas wells can be monitored remotely and in real-time thanks to IoT deployments, including temperature or pressure sensors, which improve safety, productivity, and sustainability.

## 2.2   Discussion

Table 1 summarizes the general characteristics of the above use cases, as well as the practical challenges in their implementation. The scale and dispersion of the monitoring systems vary significantly from a single field with a few sensors (as in agricultural IoT) to the European-Mediterranean region (as in seismic activity monitoring by EMSC) and

Table 1. IoT-Driven Monitoring Use Cases in Rural Environments

| Rural Environmental Monitoring Use Case | Number of Stations | Dispersion | Real-Time Constraint | Proximity to Urban Areas | Potential for Electricity Access | Potential for Internet Access | Safety Risk | Data Sensitivity |
|---|---|---|---|---|---|---|---|---|
| Air Quality (GEMS/Air) | 10s of 1000s | Global | Hour | Any | Moderate | Moderate | Moderate | Low |
| Water Quality (SWAIN) | 30 to 75 | Regional | Minute | Any | Low | Low | High | Low |
| Seismic Activity (EMSC) | $\geq 2500$ | Continental | Minute | Any | High | Moderate | High | Low |
| Avalanche (SLF IMIS) | 186 | Regional | Hour | Mid to Far | Low | Low | High | Low |
| Nuclear Explosion (CTBTO) | 337 | Global | Hour | Mid to Far | Low | Low | High | High |
| Agriculture | $\approx 1$ per 2 ha | Local | Hour | Near to Mid | Low | Low | Moderate | Low |
| Oil and Gas Well | $\approx 1$ per well | Local | Minute | Mid to Far | High | Low | High | High |

even to the whole globe with tens of thousands of sensors (as in air quality monitoring by GEMS/Air). We also observe that real-time requirements in rural environmental monitoring systems are less strict than typical latency-sensitive IoT applications such as industrial control, connected vehicles, digital twins, robotics, etc., which demand sub-second latency. The sampling frequency and available time for decision-making are both in the range of minutes to a few hours.

While some use cases, such as avalanche monitoring, are exclusively deployed in non-urban environments, others, such as water quality monitoring, include a combination of urban, suburban, and remote deployments. For instance, pollution monitoring stations in the scope of the SWAIN project can be located within cities when the river flows through them. However, other sections of the river in more rural areas have to be monitored, too. Almost all use cases require a part of the IoT infrastructure to be deployed in remote areas. Consequently, the following new challenges (CH) arise in these use cases compared to urban monitoring. An overview of all CH is presented in Figure 3.

> **CH1: Electricity Access**  In most use cases, the electric utility is unavailable at the measurement locations; therefore, the sensors have limited or no access to reliable power sources. The exceptions are (i) seismic activity sensors, which are not strictly bound to narrow geo-locations but can function in nearby settlements almost without loss of accuracy, and (ii) oil and gas well sensors since the wells are already powered.

> **CH2: Internet Access**  IoT sensors must transmit measurements to computational resources for processing. However, all rural use cases suffer from intermittent or no connectivity to a wide-area network. In theory, satellite-based communication is possible anywhere on Earth, but in practice, this solution is too costly and energy-intensive.

> **CH3: Failure Risk**  Rural deployment of IoT sensors complicates their maintenance and results in failure-prone infrastructures. Although most systems do not collect sensitive data, safety risks are generally high as failures result in undetected pollution or disasters.

> **CH4: Sustainability**  As a direct repercussion of spatially large environments and a wide dispersion of sensors, it is a challenge to achieve good coverage of the target environment. Even on relatively smaller scales, as in river or avalanche monitoring, better coverage requires a higher number of sensors, which undermines sustainability in terms of cost, energy and network use, and ecological footprint.

## 3 EDGE INTELLIGENCE

In the Edge Intelligence (EI) paradigm, a part of the computation capacity is deployed at the edge of the network, in the proximity of where the data are generated. Accordingly, the data can be at least partially processed at the edge servers, which have a high-bandwidth local area connection to the IoT devices. The output of preprocessing is usually

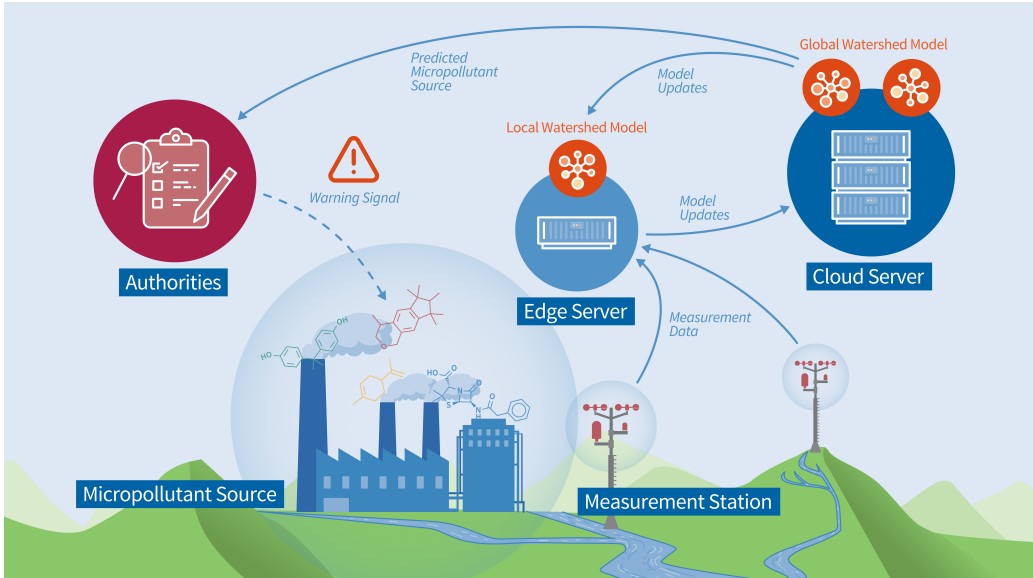

Fig. 2. Information Flow in the Edge Intelligence Architecture for Water Quality Monitoring in the Context of the SWAIN Project [12].

transferred to the central facility for further analysis [14]. Contemporary machine learning algorithms, such as deep neural networks, consist of multiple layers of processing. The size of the data that is required to be transmitted between these layers is multiple orders of magnitude smaller than the raw input data. Therefore, these layers can be partitioned between cloud and edge data centers reducing network overhead significantly and alleviating CH2. The following additional challenges have to be resolved for effective EI for environmental monitoring.

**CH5: Insularity**  Training AI models at the edge results in data scarcity and insularity. Since these models are fed with limited training data from a narrow area, the models might not generalize well. This is exacerbated by failure-prone IoT sensors (CH3) and unreliable connectivity (CH2).

**CH6: Computational Capability**  Compared to the resource-rich cloud environment, EC lacks computational capability and fan-in (the maximum number of input signals) to process streaming data from a high number of sensors, especially under energy (CH1) and cost (CH4) constraints.

**CH7: Parameter Mismatch**  There might be a mismatch between the target parameters and parameters that can be sensed due to technical limitations in sensor technology.

EI deployment in the SWAIN project is illustrated in Figure 2 to exemplify the above-listed challenges. The monitoring system consists of measurement stations equipped with various IoT sensors deployed along the river (i.e., red circles in Figure 1). Streaming data from these stations are preprocessed through local AI models in nearby deployed EC servers (for time-sensitive actions) and further analyzed in a remote resource-rich environment (e.g., cloud). Once pollution is detected, corresponding authorities are informed, along with the estimated location of the pollution source.

Here, there exist multiple EC locations considering the size of the area monitored and the lack of network connectivity. Each EC server models a part of the river and lacks a global view of the whole watershed (CH5). Therefore, a mechanism to enable them to communicate and collaborate is required. Furthermore, each EC server can only handle a limited number of measurement stations and a simplified AI model due to computational and I/O constraints (CH6). This

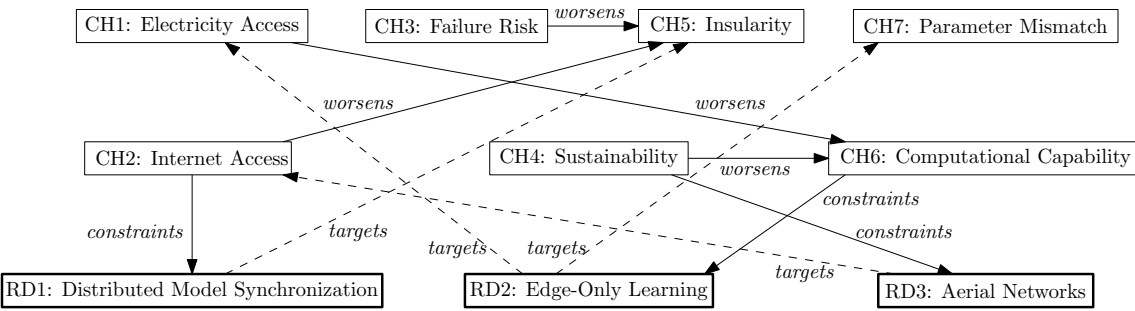

Fig. 3. The Overview of Positive (dashed arrows) and Negative (solid arrows) Relations Between Challenges (CH) and Research Directions (RD).

necessitates novel hardware with higher computational capabilities yet without higher energy consumption. Finally, the SWAIN project aims to detect micropollutants in the river; however, state-of-the-art sensors can only measure conventional water quality parameters, such as pH and turbidity (CH7). We need accurate AI-driven mapping methods to resolve the mismatch.

## 4 NEUROMORPHIC EDGE AND RESEARCH DIRECTIONS

Modern computers are almost exclusively based on von Neumann architecture, the main principles of which have remained unchanged since it was first proposed in 1945. This architecture consists of a processing unit and a separate memory that stores data temporarily during processing. The input data has to be transferred to the processing unit and the output data back to the memory through a data path. Moore's law accurately predicted the growth of the processing speed thus far. However, the steep rise in the processing unit and memory speeds started to increase pressure on the data path capacity, which stopped the already lagging growth [4].

Neuromorphic Computing [10, 11] is a new disruptive technology providing intelligent systems that imitate human neuro-biological processes through massively parallelized computing architectures. Neuromorphic hardware is not based on von Neumann architecture, as the processing unit and memory are co-located. Recently, a pioneer neuromorphic hardware developed by Intel was shown to train deep learning models in up to 81% shorter time than conventional systems [7]. Moreover, massively parallel neuromorphic circuits are event-driven. When the input signal is not present, the corresponding part of the hardware is inactive, which results in immense energy savings. Particularly in environmental monitoring with rare events of interest, NC would be highly energy-efficient. In order to increase the fan-in and data throughput (CH6) and decrease energy consumption (CH1), we propose integrating neuromorphic hardware into the edge servers. Considering the novelty of Neuromorphic EI, we identify the following future research directions. Figure 3 gives an overview of all RQ and RD, including their relations.

**RD1: Distributed Model Synchronization** Improved fan-in enables more training data and alleviates CH5. However, in geospatial applications, data from a single location might not be desirable due to non independent and identical distribution (non-IID) and the lack of variety. Therefore, local models have to intercommunicate either directly or through a parameter server. Previous work [1] demonstrates that complete synchronization is unnecessary and optimized communication can bring significant bandwidth savings. Considering CH2, it is crucial to optimize which and how much data is transmitted between edge nodes. Further research on communication-efficient, partial model synchronization is needed.

**RD2: Edge-Only Learning** Ultra-low power operation of neuromorphic hardware renders energy harvesting possible for edge computing. However, new approaches are required to optimize which data to process locally and which data to offload to remote resources (e.g., cloud). It is estimated that transmitting one bit of data requires the same amount of energy as executing 50 to 150 instructions in von Neumann computers [9]. Considering NC's improved energy efficiency and computational capability, local AI models could come into more prominence.

**RD3: Aerial Networks** Collaborative approaches between edge servers and Unmanned Aerial Vehicles (UAV), high-altitude platform stations (HAPS), or low-earth orbit (LEO) satellites [8] are promising to counter CH2 by enabling global network coverage in the future. The improved coverage through these novel technologies would benefit rural environmental monitoring systems primarily in synchronizing local model parameters. However, transmitting massive raw data without pre-processing would consume too much energy in practice. Therefore, we envision that such technologies will complement Neuromorphic EI rather than replace it, at least in rural scenarios.

## 5 NUMERICAL ANALYSIS

In this section, we focus on RD1 and present a numerical analysis that demonstrates the feasibility of model synchronization with an incomplete set of distributed models. In Figure 4, a non-IID data set in 10 partitions (y-axis) is incrementally incorporated into the model (x-axis). Here, model $n$ includes the training data from partitions $m \leq n$. Each model is then evaluated with testing data from all partitions. As expected, each model performed well with testing data from partitions that also provided training data for that model. Moreover, it is remarkable that the accuracy of the

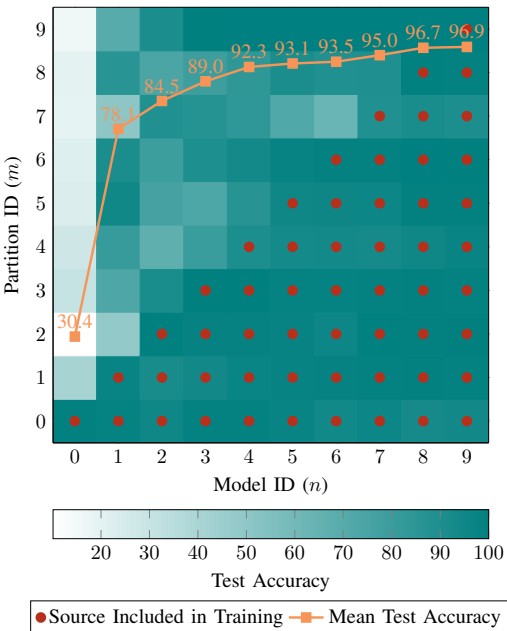

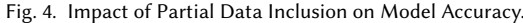

Fig. 4. Impact of Partial Data Inclusion on Model Accuracy.

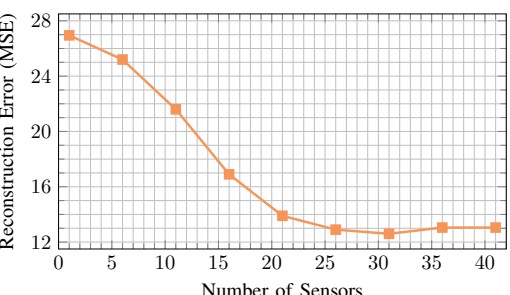

Fig. 5. Impact of Number of Sensors on Reconstruction Error in the SWAIN River Monitoring Use Case.

other partitions also improves as the model becomes more heterogeneous. The mean test accuracy line indicates that acceptable accuracy is possible with half of the partitions (i.e., $n = 4$). Figure 5 demonstrates a similar outcome from environmental monitoring. Here, 20 to 25 sensors out of 40 achieve the same performance in pollution detection. These two sets of results emphasize the potential savings in network bandwidth —a scarce resource in rural areas— utilized for synchronizing distributed models.

## 6 CONCLUSION

This paper identifies open challenges and future research directions for IoT- and AI-driven monitoring of rural environments. Furthermore, we propose Neuromorphic Edge Intelligence as a promising solution to the challenges. Among other non-von Neumann architectures, Neuromorphic Computing is arguably the most mature technology; hence researchers in this area will be the first to face the identified challenges. Therefore, we expect high interest in Neuromorphic Edge Intelligence research in the following years.

## ACKNOWLEDGMENTS

This work was supported by the grant CHIST-ERA-19-CES-005, and by the Austrian Science Fund (FWF): I 5201-N.

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
