# OpenReview forum: "Neuromorphic Edge Intelligence for Rural Environmental Monitoring"
_KDD.org/2023/Workshop/Fragile_Earth — KDD 2023 Workshop Fragile Earth Submission_

### Official Review · Reviewer_zkLD · 2023-07-11
**Review for - Neuromorphic Edge Intelligence for Rural Environmental Monitoring**

**Rating:** 7
**Confidence:** 2

**Review:**

Summary  : A discussion about the importance and challenges of Environmental Monitoring has been provided. The contents of numerical illustration provided is .However the use case of Neuromorphic Computing and its deployment to IOT and Sensors to better capture data in remote setting is quite interesting. I give it my acceptance.

Strengths :
 - Detailed and clear discussion about Environmental Monitoring, its importance and challenges has been provided.
-  Data is presented clearly.
 - Future work and concepts are explained.

Weakness :
- Issues related to Non-iid nature of data and how it is resolved is missing.
- Numerical illustration is somewhat lacking.

Questions : Has some cost analysis of deployment of Neuromorphic Architecture to remote sensors been done? What are some of the challenges in its real world deployment ?

---

### Official Review · Reviewer_b9dX · 2023-07-16
**Review for "Neuromorphic Edge Intelligence for Rural Environmental Monitoring"**

**Rating:** 7
**Confidence:** 4

**Review:**

Summary:
This paper proposes a Neuromorphic Computing as a promising solution to the energy, communication, and computation constraints in such scenarios and identify future research directions in Neuromorphic Edge Intelligence for rural environmental monitoring, namely distributed model synchronization, edge-only learning, and aerial networks. Note that, to the best of knowledge, this is the first paper to identify rural environmental monitoring as a new research direction and also the first to employ Neuromorphic EI for environmental monitoring.

Strengths :
- Novel angle of analysis.
- This paper is well written (while very dense).
- Figures are very helpful.

Weaknesses:
- More details are required in the numerical analysis.

---

### Decision · Program_Chairs · 2023-07-19

**Decision:**

Accept (Oral)

**Comment:**

Congratulations!

We are pleased to inform you that your submission: Neuromorphic Edge Intelligence for Rural Environmental Monitoring has been accepted to The KDD 2023 Workshop Fragile Earth: AI for Climate Sustainability - from Wildfire Disaster Management to Public Health and Beyond.

Camera ready deadline is ** July 24 AOE **.  Please log in to OpenReview and prepare your camera-ready version based on the reviews. Formatting rules are the same as for the initial submission and submissions must adhere to KDD 2023 guidelines available at https://authors.acm.org/proceedings/production-information/taps-production-workflow.

Again, congratulations on the acceptance of your paper!  We look forward to seeing you at the workshop on Aug 7, 2023.

The Fragile Earth Workshop Proceeding Chairs